# Spray printing of organic semiconducting single crystals

Grigorios-Panagiotis Rigas[1,2], Marcia M. Payne[3], John E. Anthony[3], Peter N. Horton[4], Fernando A. Castro[2] & Maxim Shkunov[1]

Single-crystal semiconductors have been at the forefront of scientific interest for more than 70 years, serving as the backbone of electronic devices. Inorganic single crystals are typically grown from a melt using time-consuming and energy-intensive processes. Organic semiconductor single crystals, however, can be grown using solution-based methods at room temperature in air, opening up the possibility of large-scale production of inexpensive electronics targeting applications ranging from field-effect transistors and light-emitting diodes to medical X-ray detectors. Here we demonstrate a low-cost, scalable spray-printing process to fabricate high-quality organic single crystals, based on various semiconducting small molecules on virtually any substrate by combining the advantages of antisolvent crystallization and solution shearing. The crystals' size, shape and orientation are controlled by the sheer force generated by the spray droplets' impact onto the antisolvent's surface. This method demonstrates the feasibility of a spray-on single-crystal organic electronics.

[1] Faculty of Engineering and Physical Sciences, Department of Electrical and Electronic Engineering, Advanced Technology Institute, University of Surrey, Guildford GU2 7XH, UK. [2] National Physical Laboratory, Teddington, Middlesex TW11 0LW, UK. [3] Department of Chemistry, University of Kentucky, Lexington, Kentucky 40506, USA. [4] School of Chemistry, University of Southampton, Southampton SO17 1BJ, UK. Correspondence and requests for materials should be addressed to M.S. (email: m.shkunov@surrey.ac.uk).

The progress in modern electronics in the last seven decades was closely linked with the development of single-crystalline semiconducting materials, initially represented by silicon and germanium[1], and recently enriched by a new class of organic semiconducting single crystals (OSSCs)[2–6], offering enhanced physical properties demonstrated in light-emitting transistors[7] and human tissue equivalent materials for medical X-ray detectors[8]. Organic single-crystal deposition is predominantly different from the growth of inorganic counterparts and offers the potential of low-temperature, large-scale device fabrication using solution progressing of molecular semiconductor 'inks' for high-performance, organic-printed electronic devices. Despite significant efforts devoted to the development of a scalable printing method to fabricate OSSCs[9], reported processes are either material[10,11] or substrate[12] specific, limiting widespread application.

Spray printing[13–16] of semiconductor inks is recognized as one of the best methods for achieving efficient deposition of molecular semiconductors compatible with large-area printed electronics production lines. Vertically injected spray mists have been used in the past, relying mostly on the slow evaporation rates of certain solvents to prolong the fluid to solid transition and thus improve the crystallization of the dissolved semiconducting molecules[15]. Although significant improvements have been demonstrated in the performance of field-effect transistors (FETs) by using this approach, the active layer still retains its polycrystalline nature, which is unfavourable for applications such as lasing[17] and high-energy isotope detection[18,19]. These applications require highly ordered single crystals with well-defined boundaries. Antisolvent crystallization has demonstrated its potential for controlled solidification from the solution phase of small conjugated molecules, resulting in highly ordered crystals deposited by inkjet printing[10]. However, pre-deposition patterning of the substrate was required to facilitate and confine the crystallization. In addition, when the same approach was applied for a different semiconducting molecule, it resulted in polycrystalline domains[11]. The solution shearing technique addressed the issue of one-step patterned crystallization and in some cases produced remarkable results for the charge carrier properties in FETs[20]. Unfortunately, such processes still lead to polycrystalline semiconducting films, thus making the approach incompatible with a broad range of applications.

Here we demonstrate a semiconducting organic single-crystal deposition through spray printing at an angle, conducted in air and at room temperature, coupled with solvent–antisolvent crystallization. The single-crystalline nature of the crystals has been examined by a combination of techniques probing both local molecular orientation and order with X-ray diffraction (XRD) and polarized Raman, and evaluating the overall large area uniformity of the crystals with polarized optical and scanning electron microscopy (SEM).

## Results

**Single crystals spray deposition.** In our approach, a solution of the semiconductor dissolved in a volatile solvent (hereafter referred to as the 'good solvent') is placed inside the container of a conventional aerograph, whereas a thin film of the antisolvent, in which the semiconductor molecules exhibit low or no solubility, is pre-deposited on the substrate using drop casting. We found that when a good solvent with a boiling point and surface tension lower than that of the antisolvent is used, isolated single crystals with uniform thickness distribution were grown. In addition, by optimizing the spray-printing conditions, the shearing effect of the spray mist on top of the smooth antisolvent's surface can be used for altering the shape, size and orientation of the generated crystals. In contrast with conventional spray-printing processes, our approach generates molecularly flat, isolated single crystals without the need for any pre-deposition patterning of the substrate. The majority of the results shown here were produced using a soluble pentacene semiconductor 6,13-bis (triisopropylsilylethynyl)pentacene (TIPS-PEN) synthesized with previously described techniques[21] and dissolved in toluene. However, the method is not material specific as demonstrated by the successful deposition of single crystals of a range of different organic semiconductor molecules, including anthracene, tetracene, anthradithiophene and benzothiophene derivatives (see Supplementary Figs 1–3). The antisolvent of choice was N,N-dimethylformamide (DMF), as it acts as a very poor solvent for TIPS-PEN. One of the advantages of using toluene as a good solvent is its lower boiling point ($\sim 111\,°C$) leading to fast evaporation, as compared with slow evaporating chlorobenzene ($\sim 131\,°C$)[11], which is traditionally used in organic electronics with an intention to give adequate time for the molecules to self-assemble into crystalline structures[22]. The difference with a solvent–antisolvent system is that self-assembly occurs inside the latter; thus, a rapid solvent evaporation is preferable to initiate crystallization faster. In our method, releasing the molecules promptly into the antisolvent leads to their well-defined confinement over the substrate's surface, resulting in distinct crystal structures, eliminating the need of pre-deposition patterning.

A schematic representation of the deposition process to control the crystallization of small molecules is shown in Fig. 1. A variety of substrates was used including Si with native oxide, Si with a thermally grown 230 nm $SiO_2$, glass and flexible polyethylene naphthalate. For the spray coater, two variations of an aerograph were used. The first one has an external air–paint mix system that results in a spray mist with a diameter between 6 and 25 mm at the end of the nozzle. The second has an internal mixture system, which results in finer 'atomization' of the ink and a fixed nozzle diameter of 0.2 mm. In our deposition process, the antisolvent is deposited first using a pipette (Fig. 1a) to form a uniform layer over the substrate and then the dissolved semiconductor is 'sparsely' sprayed over it. By altering the angle, distance and the type of airbrush from which the TIPS-PEN/toluene solution was ejected, we controlled the shape and the size of single crystals, with typical examples shown in Fig. 1d–g. Longer crystals were produced when a larger incidence angle (Fig. 1d) was used, whereas square-type crystals were formed at smaller angles (Fig. 1e). More specifically, when the angle was set to $\sim 75°$ (Fig. 1c) and the jetting distance was 10 cm, the crystals produced were rectangular in shape. Altering the distance between the nozzle and the substrate controlled the size of the crystals (Fig. 1f and Supplementary Fig. 4). We propose that oblique impact of the solution droplets results in a shearing mechanism (Fig. 1b) on the surface of the antisolvent, inducing patterns of semiconductor material with homogeneous distribution. Larger angles allow for the dispersion to propagate on the antisolvent's surface over a longer distance, thus forming longer crystals. Evidence of the patterning dynamics can be seen when an airbrush with finer atomization was used, resulting in nanowire-like crystals (Fig. 1g and Supplementary Fig. 5). In addition, the flow direction of the spray mist guides the orientation of the resulted crystals over the substrate (Supplementary Fig. 6). Contrary to what was observed for the toluene–DMF solvent–antisolvent system, we found that non-uniform crystalline patterns emerged (Supplementary Fig. 7a) when we used a good solvent with a higher boiling point and comparable surface tension, tetralin ($\sim 208\,°C$, $33.16\,mN\,m^{-1}$) with that of DMF ($153\,°C$, $36.2\,mN\,m^{-1}$; more details on solvent physical properties can be found in Supplementary Table 1). Our observations of crystals formation

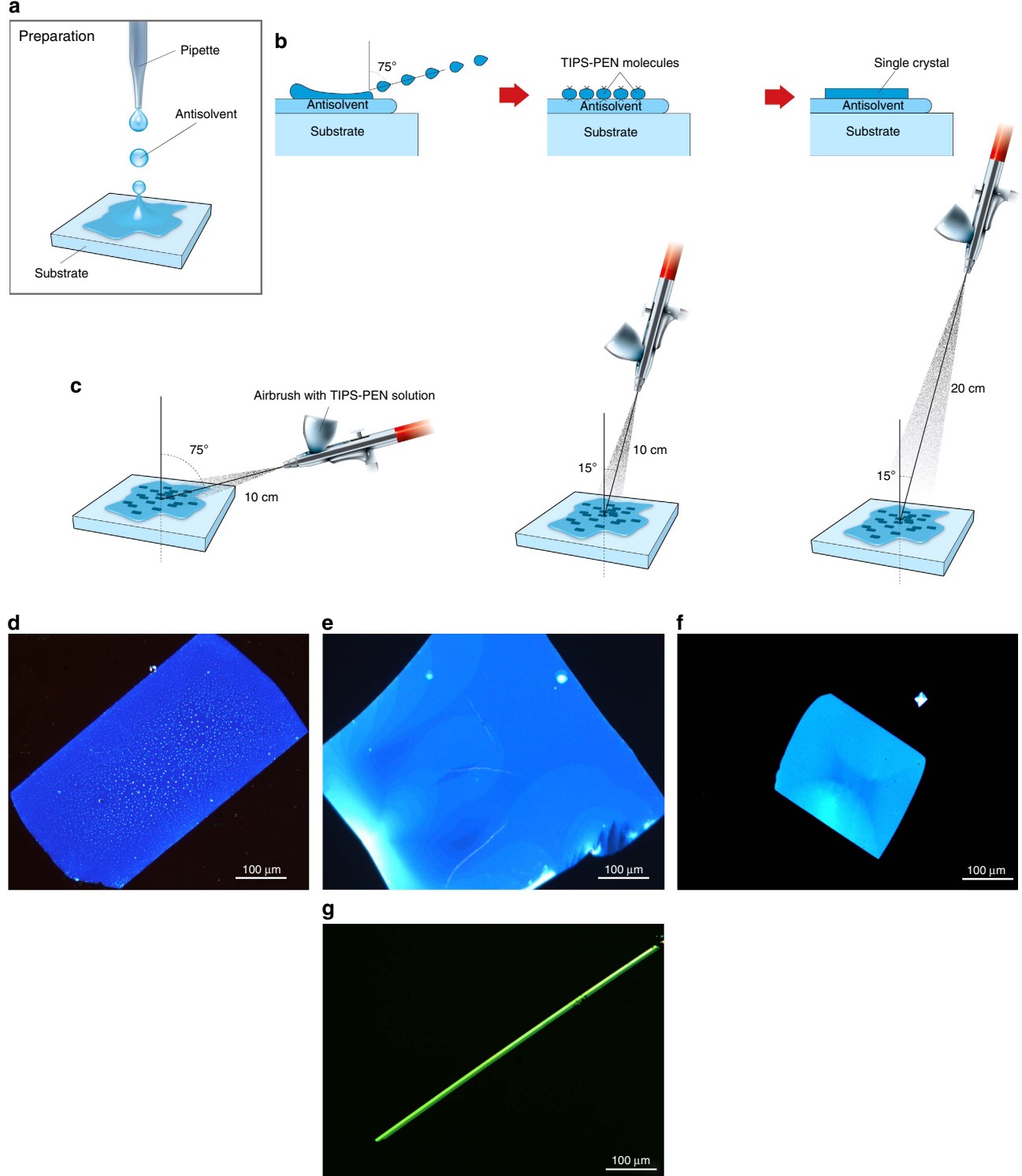

**Figure 1 | Schematic representation of experimental setup and the resulting crystal structures.** (**a**) Deposition of antisolvent. (**b**) Illustration of the proposed shearing mechanism for the droplets, generated by the airbrush, and formation of single crystals. (**c**) Schematics of three different airbrush positions influencing the shape and size of the crystals. (**d–f**) POM images of spray-printed TIPS-PEN single crystals generated under different spraying angles and distances, (**d**) 75°, 10 cm, (**e**) 15°, 10 cm and (**f**) 15°, 20 cm. (**g**) POM image of a needle-like single crystal (TIPS-PEN) made using a finer atomization airbrush. All scale bars, 100 μm.

are consistent with an earlier study on the underlying mechanisms in antisolvent crystallization for ink-jet printing[23], when miscible liquids with various surface tension values were used. We have identified that low surface tension 'good' solvent toluene (28.15 mN m$^{-1}$) leads to excellent single-

crystal formation properties on top of high surface tension DMF, in line with demonstration of good wetting behaviour of low (high) surface tension of jetted (sessile) microdroplets, respectively[23]. However, comparable surface tension liquids, tetralin and DMF, produced inhomogeneous crystallites,

indicating suboptimal growth conditions, consistent with fluid dynamic wetting–dewetting competition in microdroplets demonstrated for liquids with similar surface tension values, such as dichlorobenzene and DMF[23]. We note that solvents' boiling points also play an important part in single-crystal formation. Faster or slower evaporation of the good solvent compared with antisolvent could lead the crystallization process to initiate either at the air–solvent interface or on direct contact with the underlying substrate. Therefore, the self-assembly of semiconducting molecules into single crystals depends strongly on the drying conditions inside the solvent–antisolvent environment.

For the optimal solvent/antisolvent combination, due to the fast evaporation of toluene, DMF serves as the host medium on which the small molecules are left to self-assemble into larger domains through $\pi - \pi$ interactions[12]. The molecules that are released from individual spray droplet act as nucleation sites for their neighbouring equivalents[24], which contribute to the increase of the formed crystallite size. When the surrounding area is depleted of material (that is, no more droplets left that landed in proximity), the process is terminated and the finalized crystal is left floating on the surface of the antisolvent. The number of crystals per area varies depending on the spray-printing parameters. Typically, one crystal in an area of $350 \times 260 \, \mu m$ is grown when smaller spraying distances are used (Supplementary Fig. 6a). More than ten crystals are typically grown in the same area when larger spraying distances are used (Supplementary Fig. 6b). This is due to the large dispersion of the droplets, which are not combining to form larger patterns, leading to the formation of smaller crystals (Fig. 1f). If desirable, air flow can be used to push the floating crystal and position it on a specific substrate area (for example, on top of pre-deposited electrodes to build thin-film transistors (Supplementary Fig. 14). After their formation, crystals are floating on top of the antisolvent until it evaporates completely, letting them to settle on top of the substrate. Faster evaporating antisolvents (for example, acetonitrile; Supplementary Fig. 8 and Supplementary Table 1) can also be used with our approach for accelerating the deposition process, depending on the solubility of the active material we wish to crystallize.

**Crystal characterization**. A SEM image of a typical crystal is shown in Fig. 2b, revealing TIPS-PEN formation with well-defined boundaries and parallelogram-like shape. The crystal shape is consistent with the self-assembling packing mechanism suggested for this molecule[25]. The SEM image also shows that the crystal follows the morphology of the underlying substrate, covering any impurities on the surface. This observation verifies that the crystals were formed before the complete evaporation of DMF, followed by their subsequent deposition on the substrate. Thus, in our process the antisolvent layer acts as a protective cushion, separating the crystal formation from the morphology or type of the underlying substrate. In addition, DMF provides a smooth interface for atomically flat and uniform crystals to be formed. Despite the capability of SEM to resolve the uniformity of the overall OSSC, it lacks the ability to image grain boundaries or changes in the crystalline orientation inside the formed crystal.

Polarized optical microscopy (POM) is widely used to identify crystalline domains[26] and to assess their orientation and crystal thickness variation through the observation of interference colours[10,12,20,27]. The individual crystals demonstrate strong colour contrast due to their optical birefringence. Typical POM images in Fig. 2a show consistent colour of the crystal over its entire surface, indicating a single-crystalline domain behaviour, with very uniform distribution of the semiconducting material. In addition, when the substrate is rotated relative to the polarisers' axes, the colour change from bright to dark is continuous and homogenous across the crystal, showing the absence of crystalline boundaries or separate domains. This behaviour is typical of a highly oriented crystalline structures[12,27] (Supplementary Note 1).

For further evaluating the uniformity of the stacked molecules inside the crystals, polarized Raman spectroscopy was used[28,29]. The peaks located at 1,374 and 1,578 cm$^{-1}$ of the TIPS-PEN Raman spectrum (Fig. 2c) are associated with the vibrations across the short and long axis of the molecule, respectively[28]. When the polarization angle of the monochromatic excitation source aligns in parallel with one of the two axes of the molecule, a maximum intensity of the scattered light is observed[28,29]. As the polarization angle is rotated by 360° (Fig. 2c,d), the intensity of the modes associated with the long and short axis vibrations of the molecule increase and decrease periodically, demonstrating a strong lobe-like dependence (Fig. 2d). This highly pronounced behaviour indicates that the majority of the molecules located inside the inspected area are stacked uniformly, leading to a dependence between the polarization angle of the source and the intensity of the scattered light. This angle dependence of the Raman signal was virtually identical when the same measurements were performed at various locations across the same single-crystal sample (Supplementary Fig. 12), thus verifying the uniformity of the stacked molecules inside the grown structures. On the other hand, if the examined sample is highly polycrystalline (or isotropic) within the excitation spot size, then at all possible polarization angles the same amount of photons will be scattered for the modes in question, leaving their intensity unaffected. For verifying the aforementioned behaviour, a highly polycrystalline sample (Supplementary Note 5 and Supplementary Fig. 11a) was examined under the same conditions. As expected, the intensity of the modes in question (Fig. 2e and Supplementary Fig. 11b) remains practically unaltered, highlighting the isotropic nature of the film.

The crystal quality of the generated structures was further investigated using XRD and atomic force microscopy (AFM). XRD data for the TIPS-PEN sample demonstrate very narrow diffraction spots, as shown in Fig. 3a, indicating single-crystal behaviour. The preferential orientation of these crystals is towards the 010 (Fig. 3b) direction, which is consistent with previous observations for this type of molecule. In addition, the dimensions of the refined unit cell ($a = 7.562$ Å, $b = 7.735$ Å, $c = 16.844$ Å, $\alpha = 89.57°$, $\beta = 78.50°$, $\gamma = 83.72°$) match the known crystal structure[21] with the intermolecular distance in the order of 3.54 Å (Supplementary Fig. 10). AFM images obtained from the surface of the samples (Fig. 3c) revealed well-defined crystal terraces with a height of 1.3–1.7 nm, which is comparable to the height of a single molecular layer[20,30]. Thicker crystals can be produced by altering the solution concentration ( Supplementary Fig. 9).

**FET results**. Field-effect transistors were fabricated using both the top and bottom contact approach, to verify that the functionality of the single crystals is retained regardless of the morphology of the underlying substrate. TIPS-Pen OSSCs were grown over Si/SiO$_2$ substrates with and without predefined electrode structures (Fig. 3d,e and Supplementary Fig. 13). In both cases, the crystals retained their uniformity acting as the active medium. The resulting devices produced good FET behaviour but moderate mobility values, up to 0.4 cm$^2$ V$^{-1}$ s$^{-1}$ in the linear regime, when compared with previous reports using this molecule[20]. Comparison of absolute values of FET mobility is challenging, as it can easily be overestimated[31]. Nevertheless, our results indicate that further improvements of the FET should be possible, in both

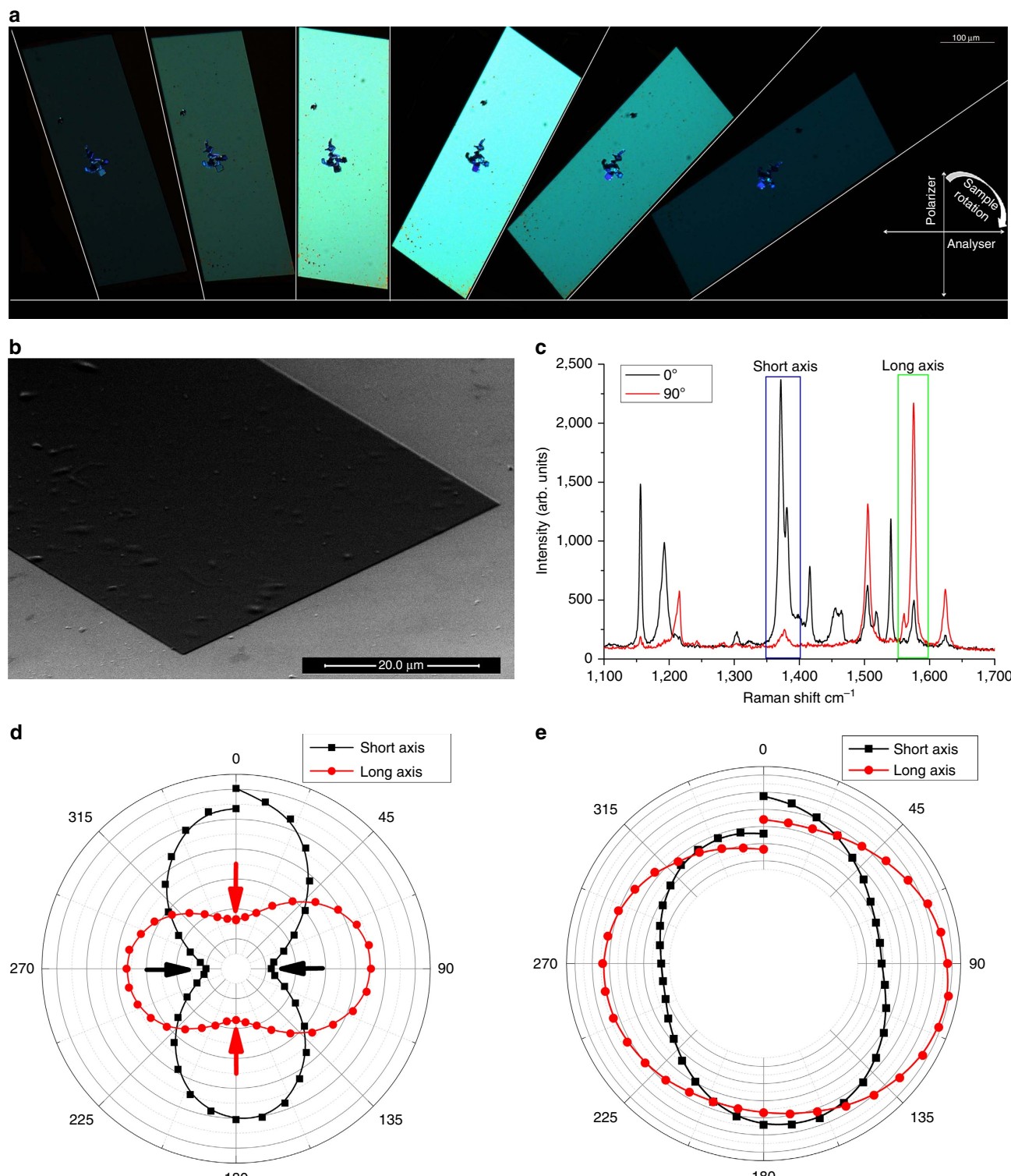

**Figure 2 | TIPS-PEN single-crystal characterization data.** (**a**) POM images of a TIPS-PEN single crystal captured under different angles. (**b**) SEM image of a TIPS-PEN single crystal after DMF evaporation. (**c**) Raman spectra captured under different polarization angles (0° and 90°), with the vibrations associated with the short and long axes of the molecule, highlighted. (**d**) Polarized Raman spectra versus angle of a single crystal. When the polarisation was rotated 360°, it demonstrated well-defined angle-dependent anisotropy of the polarized Raman spectroscopy signal. (**e**) The polarized Raman response versus angle for a highly polycrystalline sample, showing insignificant signal variations under different polarization angles.

the deposition optimization and the device processing. Both the dielectric and the electrode interfaces were not treated with self-assembled monolayers, thus leaving scope for improvement. Another limiting factor for the mobility values is possibly the intermolecular distance inside the crystal lattice. It has been previously demonstrated[20,32] that smaller intermolecular distances (down to 3.08 Å) and thus better overlap of the $\pi - \pi$ orbitals, have a significant effect in improving the charge transport of organic

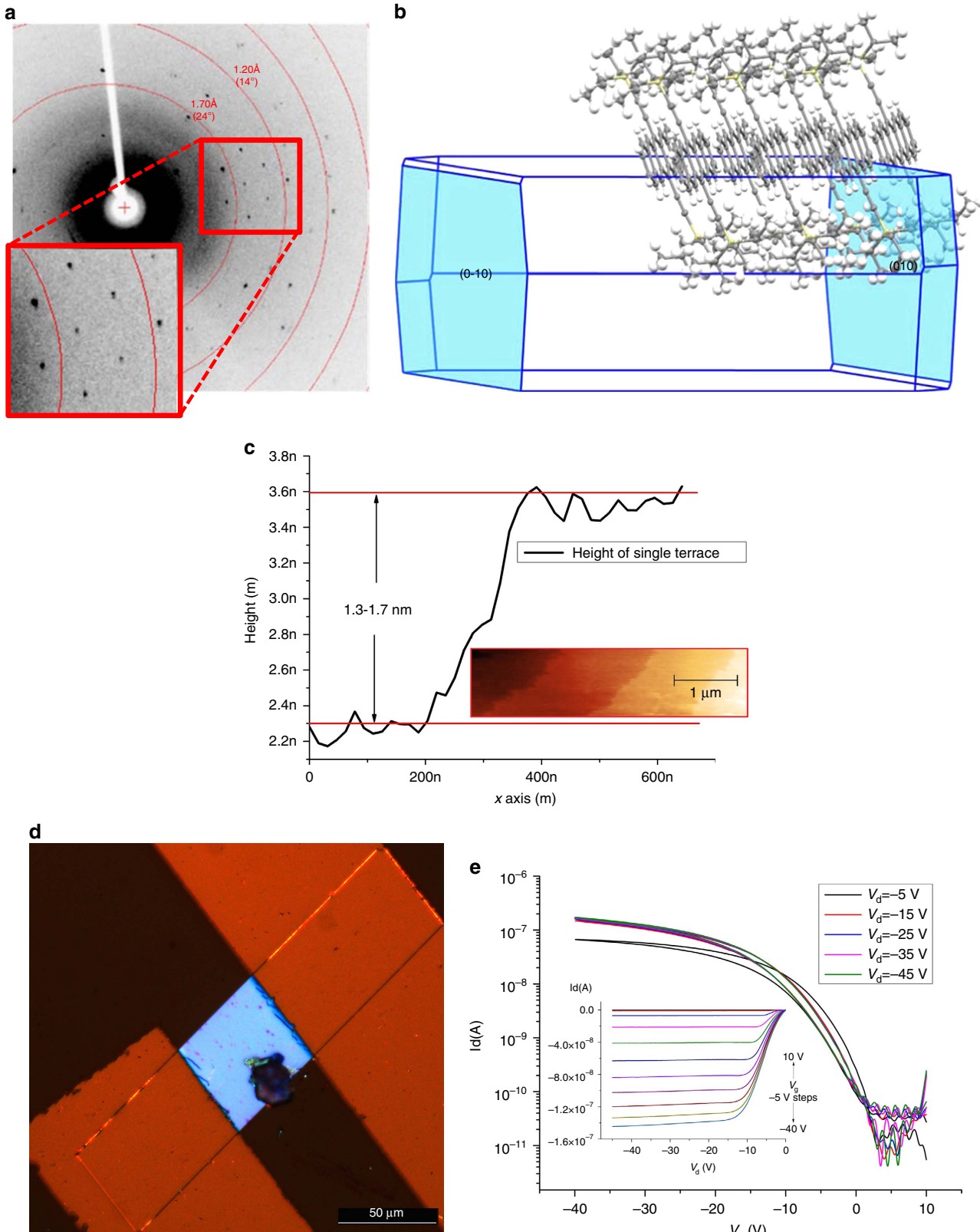

**Figure 3 | TIPS-PEN XRD and AFM characterization including transistor data.** (**a**) μ-XRD of a single crystal and (**b**) the refined unit cell with the (0–10) and (010) orientations highlighted. (**c**) AFM height profile of a monomolecular terrace. Inset AFM image showing four monomolecular terraces. (**d**) Optical microscope image of a single-crystal FET, with gold source-drain electrodes evaporated on top. The channel length is $L = 50\,\mu m$ and its width $W = 50\,\mu m$, which is equal to the width of the crystal. (**e**) Transfer and output characteristics (inset) of the device shown in **d**.

semiconductors. Further understanding and optimization of the shearing mechanism of our technique could lead to better control of the intermolecular distances, potentially leading to an increase in device performance.

## Discussion

In conclusion, the non-vacuum, room-temperature spray printing process described herein is a powerful new approach for manufacturing OSSCs with controllable shape and dimensions. The simplicity and low cost of fabricating single crystal semiconductors will facilitate the investigation of their physical properties and will provide reliable samples for the development of new measurement techniques. In addition, the method is fully compatible with a variety of molecules (Fig. 1 and Supplementary Figs 2 and 3) and it allows to generate single crystals regardless of the properties of the underlying substrate, thus paving the way for large-scale manufacturing of low-cost, organic, single-crystal printed electronics.

## Methods

**Materials and solutions preparation.** Anhydrous toluene, tetralin and DMF were purchased from Sigma-Aldrich Co., Ltd. Two milligrams of the active materials TIPS-PEN, 9,10-bis[(triisopropylsilyl)ethynyl]anthracene, 5,12-bis[(triisopropylsilyl) ethynyl]tetracene, 5,11-bis(triethylsilylethynyl)anthradithiophene and 2,7-dioctyl[1]benzothieno[3,2-b][1]benzothiophene were dissolved in 1 ml of either toluene or tetralin. All solutions were prepared inside a nitrogen-filled glovebox.

**Substrates preparation.** Both flexible (polyethylene naphthalate) and rigid (Si, SiO₂ and glass) substrates were solvent cleaned in isopropanol (IPA), ethanol and acetone, by running a 10 min ultrasonic-bath cycle for each solvent.

**Polarized optical microscopy.** We have used Leica DM2500 P polarized optical microscope coupled with a DFC420 digital camera kit. Crystals printed on thermally or natively grown oxide (SiO₂/Si) substrates were placed under the microscope's objective on a rotating sample holder. With the two polarizers crossed at $\delta = 90°$, the samples were illuminated from the top using a white light source through a variety of semi-apochromatic small working distance objectives (Leica HC PL FLUOATAR at $\times 10$ and $\times 20$ magnification, Leica N PLAN EPI $\times 100$ magnification). Owing to their optical birefringence, individual crystals show strong colour contrast. Depending on the azimuthal orientation and the thickness, the crystals appear with a specific colour. By rotating the sample holder, the anisotropy and thickness uniformity of each individual crystal was examined.

**X-ray diffraction.** A Rigaku AFC12 goniometer equipped with an enhanced sensitivity (HG) Saturn724+ detector mounted at the window of a FR-E+ SuperBright molybdenum rotating anode generator with HF Varimax optics (100 μm focus) running CrystalClear-SM Expert 3.1b27 was used. Each image was collected for 100 s as a 1° ω-scan.

**Polarized Raman spectroscopy.** A Horiba LabRAM HR coupled with a confocal microscope was used. A 785 nm laser source provided excitation for all the measurements, focused through a $\times 50$ long working distance objective. The polarization of the monochromatic light was altered using a rotating quarter wavelength plate. The Raman spectra for each of the 5° intervals was acquired. The change of the intensity of the Raman peaks, associated with the short (1,374 cm⁻¹) and long (1,578 cm⁻¹) axis vibrations of TIPS-PEN, in respect with the polarization angle was plotted using Origin 2015. The same software was used for analysing the results.

**Atomic force microscopy.** An MFP-3D (Asylum Research) AFM kit was used, using an OrcaTM cantilever holder. All measurements were conducted under ambient conditions using a platinum silicide (PtSi-FM) cantilever (NanosensorsTM) in intermittent contact (tapping) mode. The scanning frequency was set at 0.3 Hz, whereas the set point at 0.1 V for 'soft contact' of the tip with the sample. For extracting the height of the monomolecular terraces, both the trace and retrace signals were used. Gwyddion v2.39 SPM data analysis software was used for the interpretation of the results.

**Scanning electron microscopy.** A FEI Quanta 200F environmental SEM was used. Single crystals were printed on SiO₂/Si substrate and placed inside the SEM's vacuum chamber. All images were acquired at high vacuum and at 5 kV

accelerating voltage, to avoid any damage to the crystals. The samples were tilted at 50° to 70° angle for imaging the sidewalls of the structures.

**FET fabrication and characterization.** For the FET fabrication, both the bottom-contact bottom-gate (BCBG) (Supplementary Fig. 13a) and the top-contact bottom-gate (Supplementary Fig. 13b) structures were used. For the BCBG architecture, the source-drain contacts were patterned using photolithography (lift-off). Gold (Au) micro-electrodes, 50 nm thick, were deposited by sputtering (JLS MPS 500), with 4 nm Titanium (Ti) acting as an adhesion metal. Heavily n-doped Si wafers with 230 nm thermally grown SiO₂ ($C_0 = 17$ nF cm⁻²) were used as substrates. The OSSCs were then spray printed on top. No modifications of the interfaces between the OSSCs and the rest of the device components (S/D electrodes, dielectric) by using self-assembled monolayers were used. For the top-contact bottom-gate architecture, the OSSCs were spray printed on the substrate (Si/SiO₂) and the Au electrodes were then thermally evaporated on top of the crystals, through a shadow mask. Similar BCBG approach was followed for fabricating single-crystal FETs using hexagonal 2,7-dioctyl[1]benzothieno[3,2-b][1]benzothiophene crystals. The crystal was spray printed over an array of pre-patterned Au electrodes and positioned on top of the electrodes using air flow, before the evaporation of the antisolvent (Supplementary Fig. 14a). The single-crystal device produced good FET behaviour, with moderate mobility in the linear regime of 1.6 cm² V⁻¹ s⁻¹ (Supplementary Fig. 14b). No treatments of the electrodes or the dielectric were performed. For extracting the electrical response of the devices, a Keithley 4,200 semiconductor characterization system was used. To avoid atmospheric contamination, all the measurements were conducted inside a N₂-filled glovebox (H₂O < 0.3 p.p.m., O₂ < 0.1 p.p.m.). The field-effect mobility of each device was extracted using the standard MOSFET equation in the linear regime: $\mu^{linear} = \frac{L}{W\,C_0\,V_{sd}}\frac{\partial I_{sd}}{\partial V_{gs}}$, where $C_0$ is the capacitance of the oxide, $L$ is the length and $W$ is the width of the transistor channel.

**Data availability.** The data that support the findings of this study are available from the corresponding author on request.

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

## Acknowledgements

G.-P.R. and F.A.C. acknowledge funding from the UK Department for Business, Innovation and Skills. M.S. acknowledges equipment support from EPSRC grant EP/I017569/1. J.E.A. and M.M.P. thank the U.S. National Science Foundation (CMMI-1255494) for support of organic semiconductor synthesis. We thank the EPSRC for use of the UK National Crystallography Service.

## Author contributions

M.S. conceived the idea of spray-printed single crystals. G.-P.R. designed the experiments and developed the technique. G.-P.R. fabricated the samples and characterized them using AFM, SEM, POM and polarized Raman spectroscopy. G.-P.R. fabricated the FETs and electrically characterized them. G.-P.R., M.S. and F.A.C. wrote the manuscript. J.E.A., M.M.P. and F.A.C. contributed materials and analysis tools. P.N.H. conducted the XRD analysis on the samples. All authors discussed the results and commented on the manuscript.

## Additional information

**Competing financial interests:** The authors declare no competing financial interests.

**Publisher's note**: 

