## [Peer Review File · Nature Communications]

Reviewers' comments:

Reviewer #1 (Remarks to the Author):

This paper explores a method to produce single crystals of organic molecule crystals that are meant for electronic applications. The use of an antisolvent and solvent are claimed to enable the formation of these large single crystals on the surface. The main issues this paper faces to publication are:

1. Most of the analysis of single crystal structure are not so convincing. The Raman and XRD are done at one spot with micron size probe, and therefore they only tell us about the local structure. The authors would need to use 2D maps of Raman or XRD structure to show this extends across the entire region. Or they could use selected area electron diffraction taken at several locations to show the crystallographic axis remain constant. Or they could use LEED and map out the crystallography at different locations in a crystal to prove it is really a single crystals. At the moment the main evidence of uniform single crystal across the entire area is from the optical images under polarized light, but this isnt enough to confirm this.

2. The method of spraying onto DMF may lead to these single crystals as claimed, but it will make the large scale printing slower and more challenging. The authors need to demonstrate that they can print crystals in well defined arrays to then make devices. Making one device doesnt have much use, the authors need to show this method is compatible with printing scalability and make arrays of devices that prove this.

3. The paper lacks any real theoretical analysis of the mechanisms behind the formation of the proposed single crystals. There is no simulations or deeper understanding to be gained here. It reads very much as an engineering approach that occurred via trial and error, and without too much theoretical calculations that support the claims behind the physics and chemical interactions.

Overall, this paper has some interesting basic chemical synthesis, my reservations lie in the practical use of this approach in relation to speed, control and patterning demonstration. The authors need to do more work to show the crystals are single crystal using mapping methods of diffraction.

Reviewer #2 (Remarks to the Author):

This paper reports a technique of spray printing on antisolvent liquid layer for manufacturing organic semiconductor single crystals. The authors examined several printing conditions such as nozzle distance, angle, and combinations of solvents, and found that the method allows control of crystal size, shape, and orientation of the obtained single crystals without hydrophilic/hydrophobic patterning on the substrate surface. The paper is quite interesting and well written, so that I recommend the publication of this paper in Nature Communications. Prior to the publication, I would like to ask the authors to add more descriptions in the paper, by considering the following comments.

From the schematic shown in Fig. 1c, more than 10 pieces of single crystals seem to be grown on the antisolvent liquid surface. However, it is not clear from the paper how many crystals are usually grown, and what is the average distance between the crystals. The information should be important for the further study to design the process for fabricating TFT arrays.

As for the description on the dependence of solvent-solvent combinations (83rd line - 88th line in the text), I think that it might be better for the authors to consider the effect of surface tension difference

between the solvents. The details can be seen in the literature; Y. Noda et al. *Adv. Funct. Mater.* 25, 4022 (2015).

REVIEWERS' COMMENTS:

Reviewer #1 (Remarks to the Author):

The authors have addressed the comments I raised in the first review and this has led to improvements in the manuscript. I can support publication.

Reviewer #2 (Remarks to the Author):

I think that the authors fully respond to all the questions and comments by the referees. I would like to recommend the publication.

Reviewer #1

This paper explores a method to produce single crystals of organic molecule crystals that are meant for electronic applications. The use of an antisolvent and solvent are claimed to enable the formation of these large single crystals on the surface. The main issues this paper faces to publication are:

1. Most of the analysis of single crystal structure are not so convincing. The Raman and XRD are done at one spot with micron size probe, and therefore they only tell us about the local structure. The authors would need to use 2D maps of Raman or XRD structure to show this extends across the entire region. Or they could use selected area electron diffraction taken at several locations to show the crystallographic axis remain constant. Or they could use LEED and map out the crystallography at different locations in a crystal to prove it is really a single crystals. At the moment the main evidence of uniform single crystal across the entire area is from the optical images under polarized light, but this isnt enough to confirm this.

Reply:

Thank you for raising this issue. As evidence of single crystallinity we use a combination of methods including polarised optical microscopy to confirm large area crystal uniformity, and XRD and polarised Raman as molecular orientation probes. As additional supporting evidence we have now included polarised-Raman scan data acquired from multiple points over a TIPS-Pentacene crystal in the amended supplementary information file (Fig. S11). The data reveals consistent molecular orientation, which is independent of the examined location. A note for this behaviour is included in the main text.

Indeed, the question raised about the methodology to verify the anisotropy of our crystals, highlights one of the main challenges in the field of organic single crystal electronics [Ref 22 in our manuscript]. Most of the published methods (e.g. Nature 475, 364-367 (2011), Nature Materials 12, 665-671 (2013), Adv. Mater. 24, 497-502 (2012)) rely on three basic steps to support claims of single crystallinity:

- ***Generic observations of the uniformity of the crystals using polarised optical microscopy (POM) and scanning electron microscopy***
- ***Single spot diffraction patterns using XRD or TEM for more detail analysis of the crystallographic orientation and intermolecular distances.***
- ***Atomic Force Microscopy for verifying the presence of monomolecular terraces which is evidence of high anisotropy***

Typically, in the literature, the main evidence to confirm the anisotropy along the entire organic crystalline structure is based on the observations obtained from POM supported by a diffraction experiment. We go a step further and in addition to the above data (POM supported by XRD) we present 2D polarised Raman as a non-destructive technique to assess homogeneity of the crystalline structure.

Changes in the text are introduced in lines 40-43, 77-78, 137-146, 154-157, 165-166 and 170 of the main manuscript, combined with two extra references (Ref 26, 27). Additional information was included in supporting information at sections 4, 9, 11 (figures S9, S11), including POM, XRD and polarised Raman experimental setup and data.

2. The method of spraying onto DMF may lead to these single crystals as claimed, but it will make the large scale printing slower and more challenging. The authors need to demonstrate that they can print crystals in well defined arrays to then make devices. Making one device doesn't have much use, the authors need to show this method is compatible with printing scalability and make arrays of devices that prove this.

Reply:

We agree with the reviewer that printing scalability is very important. The drawback of slower evaporation (in the case of DMF) is indeed noted in the amended text and we show that other solvents can be used to accelerate the process (e.g. acetonitrile, data in Fig. S7). We also note that single crystals can be collected shortly after their formation on the top of the anti-solvent layer, thus eliminating the need to wait for complete evaporation of DMF.

As mentioned in the text, air flow or other methods can be used for transferring and positioning crystals onto specific substrates/patterned structures. We included extra data for a typical FET device produced with air flow positioning (Fig. S13) to demonstrate that the method is robust, reproducible and indeed applicable to a variety of molecules.

Demonstration of printed single crystal arrays of C8-BTBT has been included in the supporting information (Fig. S5b), as requested. The patterning potentials of our technique is evident from the controlled orientation of the crystals in respect to the substrate using only the airflow of the spray mist. This was achieved without sacrificing the control of the shape and size of the aforementioned structures. Well defined positioning of the crystals in arrays should be possible by developing a more sophisticated spray-printing equipment than the one used in this work.

The method of spray-printing onto an antisolvent layer, as described in our paper, presents several significant advantages for manufacturability of organic single crystals: independence of substrate, compatibility with a variety of molecules, controlled patterning of crystal physical dimensions, without the need of laborious and expensive photolithographic steps or hydrophilic/hydrophobic patterning on the substrate's surface.

We believe this proof of concept will stimulate further work in the field of physics of organic crystal growth for different applications, process upscaling research (ex: design the process for fabricating TFT arrays) as well as new organic single crystal devices.

Changes in the text are introduced in lines 87-88, 113-118, 120-124 of the main manuscript. Additional data was included in supporting information in sections 2 (Table S1), 6 (Fig. S5) and 12 (Fig. S13).

3. The paper lacks any real theoretical analysis of the mechanisms behind the formation of the proposed single crystals. There is no simulations or deeper understanding to be gained here. It reads very much as an engineering approach that occurred via trial and error, and without too much theoretical calculations that support the claims behind the physics and chemical interactions.

Reply:

Indeed, the focus of this communication was not a theoretical investigation, which would require a separate and long detailed study. We have aimed to communicate the important experimental findings to allow the scientific community to use the results for more in-depth studies, including

theoretical analysis of the growth mechanisms. The detailed experimental work demonstrated here was based on deep understanding of well-established mixed solvent crystallisation of organic materials which can be found in classic chemistry textbooks (ex: Crystal Growth of Organic Materials, edited by Myerson, Green, and Meenan, ACS Proceedings Series, 1996) as well as detailed knowledge of a variety of inspiring publications (e.g. Nature 475, 364-367 (2011), Adv.Mater. 24, 497-502 (2012), Adv. Funct. Mater. 25, 4022-4031 (2015)). These basic principles were coupled with our experimental expertise and the results were thoroughly correlated using standard practices in the field.

We agree with the reviewer in a sense that, due to its novelty, this work will trigger a broad theoretical work in areas such as droplet impact dynamics, design of small molecules for improved crystallisation, self-assembly mechanisms for small molecules, charge carrier transport in solution processed single crystals etc. The main criticism to most of the theoretical work that is being published, is the lack of experimental proofs for the theoretical claims. To the best of our knowledge, our technique for producing well defined organic single crystals from a variety of materials, is the simplest available in literature. This will benefit research groups with limited experimental base to support their theoretical findings. Due to its low deposition equipment cost, the technique can be accessible to various research groups which may lack the necessary resources for more sophisticated equipment.

Overall, this paper has some interesting basic chemical synthesis,

- my reservations lie in the practical use of this approach in relation to speed, control and patterning demonstration.

Reply:

We answer this question in comment #2 and the extra data were included in supporting information (Figs. S5, S13)

- The authors need to do more work to show the crystals are single crystal using mapping methods of diffraction.

Reply:

Answered in comment #1 and the extra data were included in supporting information as requested (Figs. S11).

Reviewer #2 (Remarks to the Author):

This paper reports a technique of spray printing on antisolvent liquid layer for manufacturing organic semiconductor single crystals. The authors examined several printing conditions such as nozzle distance, angle, and combinations of solvents, and found that the method allows control of crystal size, shape, and orientation of the obtained single crystals without hydrophilic/hydrophobic patterning on the substrate surface. The paper is quite interesting and well written, so that I recommend the publication of this paper in Nature Communications. Prior to the publication, I would like to ask the authors to add more descriptions in the paper, by considering the following comments.

From the schematic shown in Fig. 1c, more than 10 pieces of single crystals seem to be grown on the antisolvent liquid surface. However, it is not clear from the paper how many crystals are usually grown, and what is the average distance between the crystals. The information should be important for the further study to design the process for fabricating TFT arrays.

Reply:

The schematic in Fig. 1c was meant to serve as an indication of the process. We included more experimental data in the amended supporting information (Fig. S5) to give more examples of multiple, oriented crystals formation. We also included a description in the main text as follows: "The number of crystals per area varies depending on the spray-printing parameters. Typically, one crystal in an area of 350x260µm is grown when smaller spraying distances are used (Fig. S5a). More than 10 crystals are typically grown in the same area, when larger spraying distances are used (Fig. S5b). This is due to the large dispersion of the droplets which are not combining to form larger patterns, leading to the formation of smaller crystals."

We believe, that development of a more sophisticated spraying equipment is the next logical step for fabricating large TFT arrays. Such an equipment should produce a more uniform distribution of the droplets over the antisolvent and, thus, smaller discrepancies in the distances between the crystals.

Changes in the text are introduced in lines 113-118 of the main manuscript, and in supporting information in section 6, Fig. S5.

As for the description on the dependence of solvent-solvent combinations (83rd line - 88th line in the text), I think that it might be better for the authors to consider the effect of surface tension difference between the solvents. The details can be seen in the literature; Y. Noda et al. Adv. Funct. Mater. 25, 4022 (2015).

Reply:

We would like to thank the reviewer for bringing this inspiring article to our attention. Indeed surface tension between different solvents appears to have significant effect in the spreading of the semiconducting molecule formulation over the antisolvent's surface, and the initial contact dynamics of the spray microdroplets on the antisolvent surface is particularly important. We note that in our printing process the volume of deposited microdroplets is significantly smaller than the volume of antisolvent layer on the substrate. Additionally, the surface tension of 'good' solvent

(toluene, 28.5mN/m) is noticeably lower than that of antisolvent DMF (36.2mN/m). Thus, droplets dynamic is being fully consistent with 'wetting' behaviour of two chemically different microdroplets described by Y. Noda et al. (Adv. Funct. Mater. 25, 4022, 2015). However, we anticipate that the angle of incidence of the droplets might influence the initial solvent/antisolvent interfacial behaviour, probably enhancing the surface tension differences effect between two liquids.

Changes in the text are introduced in lines 47-48, 91-102 of the main manuscript, and in supporting information: section 2, Table S1.

REVIEWERS' COMMENTS:

Reviewer #1 (Remarks to the Author):

The authors have addressed the comments I raised in the first review and this has led to improvements in the manuscript. I can support publication.

Reviewer #2 (Remarks to the Author):

I think that the authors fully respond to all the questions and comments by the referees. I would like to recommend the publication.

Our reply: We thank the reviews for their consideration of this work, and we note that no changes are required at this round of reviews.